# Design of a randomised controlled trial: does indirect calorimetry energy information influence weight loss in obesity?

Jonathan Lim [iD] ,[1,2] Uazman Alam,[1,2] Daniel Cuthbertson,[1,2] John Wilding[1,2]

► Prepublication history and additional materials for this paper is available online. To view these files, please visit the journal online (http://dx.doi.org/10.1136/bmjopen-2020-044519).

[1]Department of Cardiovascular and Metabolic Medicine, University of Liverpool Institute of Ageing and Chronic Disease, Liverpool, UK
[2]Department of Diabetes & Endocrinology, University Hospital Aintree, Liverpool, UK

**Correspondence to**
Dr Jonathan Lim;
j.z.lim@liverpool.ac.uk

## ABSTRACT

**Introduction** Respiratory quotient (RQ) provides an indication of the relative balance of carbohydrate and fat oxidation. RQ could serve as an early biomarker of negative energy balance during weight loss. Restriction of energy intake relative to total daily energy requirements produces a negative energy balance which can lead to a fall in RQ, accompanied by a decrease in resting energy expenditure (REE). However, the net change in body weight does not usually match predicted weight change due to intraindividual metabolic adaptations. Our aim is to determine the effectiveness of utilising EE information from indirect calorimetry during weight loss intervention.

**Methods and analysis** We will undertake an assessor-blinded, parallel-group randomised controlled trial of 105 adults with obesity randomised in 1:1 ratio to receive either standard weight management care (SC) or EE information plus SC (INT) during a 24-week multicomponent weight management programme. The primary outcome is difference in weight loss between INT and SC group at 24 weeks. Secondary outcomes include: change in RQ, REE, glycaemic variability, and appetite-relating gut hormones (glucagon-like peptide 1, gastric inhibitory polypeptide, peptide YY). Generalised linear mixed models (intention to treat) will assess outcomes for treatment (INT vs SC), time (baseline, 24 weeks) and the treatment-by-time interaction. This will be the first study to evaluate impact of utilising measured REE and RQ on the lifestyle-based intensive intervention programme.

**Ethics and dissemination** Ethics approval was obtained from the Health Research Authority and the North West Research Ethics Committee (18/NW/0645). Results from this trial will be disseminated through publication in peer-reviewed journals, national and international presentations.

**Trial registration numbers** NCT03638895; UoL001379.

## INTRODUCTION

The prevalence of obesity has reached pandemic levels both worldwide and in the UK, having more than doubled globally over the last three decades.[1] Reducing the obesity-related health burden as well as tackling the obesity pandemic is core to WHO 'Global Action Plan for Prevention and Control of Non-Communicable Diseases 2013–2020'.[2] The obesogenic environment is

### Strengths and limitations of this study

► This will be the first randomised controlled trial to evaluate whether providing energy expenditure (EE) information during interactions between dietician and participants can influence weight loss during a weight loss intervention in people with obesity.

► Exploratory information on a broad range of outcomes will be evaluated including but not limited to change in body weight, body composition, resting EE and substrate oxidation which will inform the design of future weight management trials.

► The results of study will advance our current understanding of the impact of utilising EE information to inform or predict body weight change and substrate oxidation in people with obesity.

► A limitation of the study design is that the indirect calorimetry only provides information about the EE based on the time-constraints of those settings (fasted and resting state) only which may not reflect day-to-day variations in EE due to influence of energy intake and physical activity.

► While the ECAL indirect calorimetry is a portable device and more convenient to use than the ventilated hood systems, it has relatively wide limits of agreement and degree of variance between repeated measures of resting metabolic rate, respiratory quotient and substrate oxidation and thus may be less informative than more precise measures of EE.

a consequence of industrialisation of labour and food systems, leading to overconsumption of energy-dense, nutrient-poor foods combined with an increasingly sedentary lifestyle.[3] Behaviour-centred weight loss through intensive lifestyle modifications has proven effective in the short-term in attaining modest but clinically significant weight loss.[4 5] However, excessive energy intake over prolonged period leads to inability to maintain the consistent negative energy deficit required to achieve weight loss. Individuals seeking weight loss encounter much difficulty due to increased hunger and a

disproportionate reduction in energy expenditure (EE) driven by neurohormonal adaptations.[6] Consequently, maintenance of weight loss is difficult and weight regain common.[4 7]

Practical methods to better engage with people using information, self-directed education and promote effective weight loss are continually evolving among specialist weight management programmes. The ECAL indirect calorimeter (IC) (Metabolic Health Solutions, Perth, Australia) is an open-circuit portable device that measures fractional volume exchange in oxygen consumed ($VO_2$) and carbon dioxide expired ($VCO_2$) within a small mixing chamber.[8] The ECAL IC uses a proprietary mouth piece and nose clip. $O_2$ measurements were calculated with a galvanic fuel cell oxygen sensor and $CO_2$ measurements were obtained using a novel light emitting diode nondispersive infrared gas sensor. $VO_2$ and $VCO_2$ gas exchange measurements were derived from physiological methods based on Weir's equation.[9] Data generated from breath-by-breath exchange during a steady volumetric flow while in an energy-balanced resting state, generates the resting EE (REE) and respiratory quotient (RQ). ECAL IC has been compared against other IC including the GEM and DeltaTrac. Kennedy *et al*[8] reported that measures of $VO_2$, resting metabolic rate (RMR), RQ, carbohydrate oxidation, and fat oxidation showed greatest variation on the ECAL IC as compared against Deltatrac (as the standard reference device). The mean difference in RMR measures collected on ECAL and Deltatrac showed wide limits of agreement (lower 95% limit of agreement was −2562 kJ/day and upper 95% limit of agreement was 3480 kJ/day). A greater proportional bias was observed between measured RMR of ECAL against Deltatrac suggesting that at higher RMR the difference between the two devices was greater but was acceptable for repeat RMR measures between individuals. In terms of repeatability of measures of ECAL IC (intramachine variability), Kennedy *et al* reported that no significant differences were found between repeated measures of $VCO_2$, RMR, RQ and substrate oxidation measures. However, greatest bias occurred within the ECAL with a mean difference of 475±1083 kJ/day and wide limits of agreement (−2641 and 1691 kJ/day). Comparing between devices, coefficient of variance was 4 (±5.3)% on the Deltatrac, 4.9 (±4.5)% on the GEM and 11.2 (±12.1)% on the ECAL.[8] Deltatrac was previously considered the reference standard after validation studies demonstrated low bias and good precision in comparison to Douglas bag method, however, the Deltatrac is now discontinued from commercial sale.[10] The RQ is the index of $VCO_2$ against $VO_2$ which is expressed as a ratio. Typically, metabolism consisting solely of fat or lipids generates an RQ of 0.70–0.75. The approximate RQ of mixed metabolism of fat and carbohydrate is 0.80. Likewise, protein metabolism generates approximate RQ of 0.80. Metabolism consisting solely of carbohydrate produces RQ of 1.0.

Longitudinal prospective studies reported the association between low RQ (<0.75) with weight loss and weight loss maintenance.[11–14] Numerous epidemiological and experimental studies further support this correlation that a lower RQ predicts weight loss.[12 15–17] Goris and Westerterp reported postabsorptive RQ was strongly related to change in body mass (r=0.57; p=0.0001), but not body mass index (BMI), fat mass or fat-free mass (FFM).[18] Conversely, adjusting for variables of sex, ethnicity, familial predisposition, those with higher RQ were associated with weight gain or struggled to maintain the loss.[11 15] In the Baltimore Longitudinal study in Ageing, Seidell *et al* reported that high fasting RQ was a weak but significant predictor of weight gain.[15] In previously overweight and obese subjects, high RQ predicted an increase in body fat mass independent of energy balance, circulating insulin and insulin sensitivity.[19 20] Marra *et al* reported baseline RQ in non-obese women was a significant predictor of body weight (p<0.05) after a 6-year follow-up period.[20] Hainer *et al* reported that people with obesity on very low calorie diet with high RQ regained weight at the 2-year follow-up.[21] Based on the current evidence in literature, RQ serves as a valuable indicator of substrate oxidation during an energy-balanced state. If substrate oxidation was examined in a positive energy balance state, that is, energy intake >expenditure, typically carbohydrate oxidation increases and fat oxidation decreases.[22] Conversely, if measured in a negative energy balance state, that is, energy intake <expenditure, carbohydrate oxidation decreases and fat oxidation increases.[23] The predominant factors that influences the accuracy of RQ measure of substrate oxidation is determined by body composition, energy intake and EE.[24]

Behavioural weight loss programmes classically involve decreased calorie intake, increased EE and use behavioural strategies such as goal setting and self-monitoring. This is the first study that has been specifically designed to investigate the utilisation of EE information (REE and RQ) from the ECAL IC on the change in body weight. We postulate use of individualised EE information generated from the portable ECAL IC will result in greater weight change behaviour modification.

This paper describes the protocol and statistical analysis plan for the randomised controlled trial of EE from ECAL IC in a Multicomponent Weight Management Service (NCT03638895). This RCT incorporates the use of EE information from the ECAL IC vs standard care (SC) in participants with obesity and severe obesity attending a secondary care based specialist weight management service (SWMS).

## OBJECTIVES
### Primary objective
The primary aim of the study is to determine the effectiveness of providing EE information from ECAL IC to influence the outcome of weight loss in obese individuals without diabetes receiving dietary restriction, exercise and behaviour modification therapy.

## Secondary objectives

The secondary objectives are to compare the health outcomes in terms of: (1) association of weight change with the change in measured RQ; (2) association of weight change with change in measured REE; (3) change in glycaemic variability; (4) change in appetite-regulating gut hormones (glucagon-like peptide 1 (GLP-1), gastric inhibitory polypeptide (GIP) and peptide YY (PYY).

## METHODS AND ANALYSIS

### Study design

Study design is a 24-week parallel-group randomised controlled trial. The study protocol V.1.3 (Date: 28 August 2020) has been approved by the Health Research Authority and the North West Research Ethics Committee (REC 18/NW/0645). The design, conduct and reporting of this study has been approved and sponsored by the University of Liverpool/Liverpool Joint Research Office.

### Patient and public involvement

Patients were involved in the design and conduct of this research. During the feasibility stage, priority of the research question, choice of outcome measures, and methods of recruitment were informed by discussions with patients through two focus group sessions. Members of the Tier 3 SWMS at Liverpool University Hospitals National Health Service (NHS) Trust also identified this research as being a priority area for clinicians and patients undergoing weight management intervention (INT). Once the trial has been published, participants will be informed of the results through the local trust website and will be sent details of the results in a study newsletter suitable for a layperson audience.

### Trial oversight and governance

The study sponsor is the University of Liverpool and the study is managed and overseen by the University of Liverpool/Liverpool Joint Research Office. The study is registered at the http://www.clinicaltrials.gov (NCT03638895) before the enrolment of the first participant and monitored by the University of Liverpool/Liverpool Joint Research Office. Safety of the participants will be monitored by the University of Liverpool/Liverpool Joint Research Office. Decision for any interim data analysis and stopping guidelines will be assessed and validated through the University of Liverpool/Liverpool Joint Research Office.

### Participants

#### Eligibility criteria

To determine eligibility, the participants must fulfil all the eligibility criteria (box 1). Criteria were designed to ensure that participants were able to safely engage with the weight management programme.

### Recruitment

Recruitment of participants will take place from February 2019 to February 2022 in Liverpool University Hospitals

---

### Box 1 Inclusion and exclusion criteria from protocol

**Inclusion criteria**
- Man or woman, 18–70 years of age.
- BMI $\geq$30 kg/m$^2$ to $\leq$60 kg/m$^2$ at screening visit.
- Stable weight (change of <5% within 12 weeks before screening based on medical history).
- Subjects are in the investigators opinion, well-motivated, capable and willing to learn how to undergo IC testing, as required for study.
- Willing and able to adhere to the prohibitions and restrictions specified within this protocol.

**Exclusion criteria**
- Taking weight loss medication within 12 weeks prior to randomisation.
- Previous or planned bariatric surgery.
- History of type 1, type 2 diabetes mellitus, diabetic ketoacidosis, or diabetes secondary to pancreatitis.
- Has a HbA1c of $\geq$6.5% (or $\geq$48 mmol/mol).
- History of obesity with a known secondary cause (eg, Cushing's disease/syndrome).
- Oral corticosteroid use (except in the short-term use of a course of 7–10 days).
- Ongoing, inadequately controlled thyroid disorder defined as thyroid-stimulating hormone >6 mIU/L or <0.4 mIU/L.
- History of malignancy within 3 years before screening (or diagnosis of malignancy within this period) eGFR $\leq$30 mL/min/1.73 m$^2$ on serum testing.
- Alanine aminotransferase level is >2.0 times the upper limit of normal or total bilirubin is >1.5 times the upper limit of normal at screening.
- Other major illness likely to preclude participation in the trial.
- History of glucagonoma.
- A myocardial infarction, unstable angina, revascularisation procedure (stent or bypass graft surgery) or cerebrovascular accident within 12 weeks before screening.
- Heart failure NYHA class III–IV.
- End-stage chronic obstructive pulmonary disease.

**Withdrawal criteria**
- Terminal illness or loss of capacity during participation in clinical trial.
- In the opinion of the investigator is unsafe for continuation in study for medical, safety, regulatory or other reasons
- Lost to follow-up.
- Female participants with a positive pregnancy test.

BMI, body mass index; eGFR, estimated glomerular filtration rate; HbA1c, glycosylated haemoglobin; IC, indirect calorimeter; NYHA, New York Heart Association.

---

NHS Foundation Trust, Liverpool, UK. Participants will be recruited through current referrals to Aintree LOSS community-based weight management service, face-to-face clinic encounters, group-based education sessions, electronic database of participants attending the tier 3 SWMS, and notices in the hospital. Participants with BMIs $\geq$30 kg/m$^2$ will be identified from referral lists and weight management clinics. Participants will receive a study letter which briefly explains study aims and advises that researchers will be in contact within 2 weeks to provide further details of the study. If participants do not respond

to study advertising within 6 months of commencement of recruitment, there will be a further broadening of recruitment sites to include local weight management services subject to prior ethical approval. All individuals identified will be given a patient information leaflet and required to provide written informed consent prior to enrolment. People interested in participating will be met by researchers at the study site to obtain consent and complete the baseline questionnaires.

## Randomisation, allocation concealment and sequence generation

Data collected from the screening visit will be used to assign participants to groups based on sex and BMI in the process of randomisation by minimisation. This process will ensure the baseline characteristics are balanced between the treatment groups and have been used as a reasonable method for randomisation for small clinical trials to reduce bias. Participants will be randomised at an individual level by an independent statistician without contact with participants during the trial. Using a computer-based random number-producing algorithm, allocation sequence within will be generated to allow equal ratio of 1:1 to either the INT or control group. Complete separation will be achieved between the statistician who generated the randomisation sequence and those involved in assessing participants and performing data entry.

It will not be possible to 'mask' researchers or participants to group allocation. However, those responsible for analysis will be masked to group allocation. A research assistant who will not be involved in the enrolment, assessment or allocation of participants will pack and sequence prepacked envelopes with the group allocation. Only after the study investigator/subinvestigator reviews the eligibility based on health records, laboratory results and concomitant medications, and confirms eligibility to proceed, will the envelope be opened and details of the particular study group will be revealed to the participant. If the participant is randomised into the INT group, layperson client printouts and summary will then be provided with their programme resources.

## Sample size calculation

The sample size calculation was based on retrospective dataset from the SWMS,[25] with a derived SD of 4 kg. Thus, 42 participants in each group will give our study 80% power to detect a difference in weight change between groups of 3 kg at the 5% significance level using a two-sided test. A between-group difference of 3 kg was chosen as this is outside the range of normal weight fluctuation and is sufficient to sustain clinically meaningful weight loss.[5] Accounting for attrition rate of 20%, the total target recruitment is 105 participants.

## Study INT

The INT group participants will receive EE information generated from ECAL IC which will encompass the

REE and RQ delivered in the form of a layperson client report containing summary of recommendations to help improve understanding about metabolic health (example of client report can be found in online supplemental appendix. During the study visit, these results will serve as a reference tool for the dietitian when formulating a dietary plan based on the measured REE to determine total daily EE (TDEE). Further, the RQ data will also be used as an indicator of substrate oxidation to deliver key messages on carbohydrate vs fat oxidation and to facilitate decision making in energy-restricted dietary recommendations. The clinician will explain the measured REE (supposedly more accurate than predicted REE) in the context of weight loss and make recommendations on dietary plan for example, energy restrictions based on measured REE. Food diaries will be provided to study participants to support and check compliance and used to record compliance with recommendation. The threshold of compliance with study INT is set at those who complete at least five out of the nine study visits from baseline to final visit with completion of the IC test. The energy requirements of each participant in the INT group will be calculated using the measured REE and the self-reported physical activity captured via the International Physical Activity Questionnaire (IPAQ). The recommended energy restriction for weight loss will be up to 30% less than total daily energy requirement to achieve the intended weight loss. The individualised weight loss plans will be modified at study visits based on the REE and RQ information and the weight loss plans adapted if more energy restriction is required. The dietitians will use standardised diet checklists to assist with monitoring dietary compliance and cross-check with the self-completed food diaries. We have set the threshold of compliance with dietary recommendations at 50% compliance.

In comparison, the participants in the SC or control group will receive usual care delivered via multidisciplinary weight management INT including dietary INT, recommendations of physical activity and behaviour modification. Dietary restrictions will be estimated based on the Harris-Benedict equation for total daily energy requirements, with a recommended 30% energy deficit from total daily energy requirement. Typically, suggested meal plans range from approximately 1200–2500 kcal per day. Dietetic advice will be based on the Association of UK dieticians (BDA)[26] recommendation on daily intake of carbohydrates (40%–60%), protein (20%–30%) and fat (20%–30%). Participants will be encouraged not to have prolonged episodes of fasting, but to maintain up to three regular portioned balanced meals/day, while limiting snacks and reinforcing portion control. Dietitian will provide the participant with a food diary and offer them self-monitoring tools to monitor diet using MyFitnessPal Mobile App (www.myfitnesspal.com) for at least 3 days per week. Up to 150 min of physical activity/week will be recommended. Participants will be encouraged to formulate personalised dietary restriction and achievable activity goals each week during each study visit. In order to make a comparison of indirect

**Table 1** Assessments and procedures during study visits

| Procedure visit window (days) | SCR ±7 | W0 | W1 ±3 | W2 ±3 | W4 ±3 | W8 ±3 | W12 ±3 | W16 ±3 | W20 ±3 | W24 ±3 |
|---|---|---|---|---|---|---|---|---|---|---|
| Visit | 1 | 2 | 3 | 4 | 5 | 6 | 7 | 8 | 9 | 10 |
| Informed consent | X | | | | | | | | | |
| Inclusion exclusion criteria | X | X | | | | | | | | |
| Randomisation criteria and randomisation | | X | | | | | | | | |
| Medical history | X | | | | | | | | | |
| Physical examination | X | | | | | | | | | |
| Indirect calorimetry | | X | X | X | X | X | X | X | X | X |
| Full blood count | X | | | | | | | | | |
| Renal profile | X | | | | | | | | | |
| Lipid profile | | X | | | | | | | | X |
| Liver function test | X | | | | | | | | | X |
| Thyroid function | X | | | | | | | | | |
| GLP1, GIP, PYY | | X | | | | | | | | X |
| Fasting glucose | | X | | | | | | | | X |
| HbA1c | X | | | | | | | | | X |
| Urine pregnancy test | X | | | | | | | | | |
| Height | X | | | | | | | | | |
| Weight | X | X | X | X | X | X | X | X | X | X |
| Waist, hip, thigh circumference | | X | | X | X | X | X | X | X | X |
| FFM, FM | | X | | X | X | X | X | X | X | X |
| Blood pressure | X | X | | | X | X | X | X | X | X |
| Diet counselling | | X | | | X | X | X | X | X | X |
| Food diary | | X | X | X | X | X | X | X | X | X |
| Compliance check | | | X | X | X | X | X | X | X | X |
| IPAQ | | X | | | | | X | | | X |
| Meal test | | X | | | | | | | | X |
| CGM | | X | | | X | | X | | | X |
| Concomitant medication check | X | X | X | X | X | X | X | X | X | X |
| AE reporting | | X | | | X | X | X | X | X | X |
| KOSC | X | | | | | | | | | X |
| ESS | X | | | | | | | | | X |
| STOP-BANG | X | | | | | | | | | X |
| IWQoL-Lite | | X | | | | | | | | X |
| EuroQoL-5D-5L | | X | | | | | | | | X |
| Acceptability survey | | X | | | | | | | | X |

W0, week 0; W1, week 1; W2, week 2; W4, week 4; W8, week 8; W12, week 12; W16, week 16; W20, week 20; W24, week 24.
AE, adverse events; CGM, Continuous Glucose Monitor; EQ-5D-5L, EuroQoL-5 Dimension-5 Level; ESS, Epworth sleep score; ESS, Epworth Sleep Score; FFM, fat-free mass; FM, fat mass; GIP, gastric inhibitory polypeptide; GLP-1, glucagon-like peptide 1; HbA1c, glycated haemoglobin; IPAQ, International Physical Activity Questionnaire; IWQoL, Lite- Questionnaire for quality of life; KOSC, King's obesity staging criteria; PYY, peptide YY; SCR, screening visit; STOP-BANG, screening questionnaire for obstructive sleep apnoea.

measurements of REE and RQ between groups, the participants receiving SC will also undergo IC measurement, but without receiving the EE information. For full details on assessments during study visits, refer to table 1.

## Study outcomes
### Primary outcome

The primary outcome is the difference in change of weight, in absolute value (kg), between participants in the INT group (EE information plus SC) versus the SC group at baseline and 24 weeks after randomisation.

## Secondary measurements and outcomes

Anthropometrics, weight change, body composition (fat mass, FFM), obesity-associated comorbidities, changes in REE, RQ and substrate (fat and glucose) oxidation based on IC, will be measured and determined throughout the study.

### Data collection

The following section outlines the data and biochemical evaluations being collected during the test periods (see table 1 for a summary).

### Anthropometry

Assessment of anthropometric data will be obtained at screening, baseline, week 4, 8, 12, 16, 20 and 24 visits. Body weight will be measured and recorded to the nearest 100 g following an overnight fast and will be captured twice on each occasion. The weighing scale used will be the same scale which will be calibrated and used throughout the study. Height will be measured twice to the nearest 1 mm with the average value taken and recorded from a stadio metre at the screening visit. BMI will be determined as weight/height squared ($kg/m^2$). Body composition will be determined using the two-electrode leg-to-leg bioimpedance analyser machine (Tanita TBF-300MA, Tanita, Tokyo, Japan) and the average value calculated. Participants will wear a light gown, and all external metal objects will be removed prior to measurement. Total body fat mass (%, kg), total body lean mass (%, kg) will be obtained. Waist circumference will be measured to the nearest 1 mm, using a measuring tape at the midpoint between the lower costal border and the iliac crest with the average of 3 measurements taken. In addition to the baseline anthropometric measurements and end of weight loss INT (week 24), weight will also be recorded during dietetic visits and counselling to provide feedback to participants in both groups. Regular weight monitoring allows the study clinicians and participants to assist with weight monitoring, associated with improved success and compliance to weight loss INT.

### Biochemical measures

At screening, baseline, and end of study (week 24) visits, fasting (>8 hours) venous blood samples will be collected by study clinician. Collected blood samples will be centrifuged (at 4°C, 4000 rpm, 10 min) to separate plasma or serum and stored at −80°C for subsequent analysis.

### ECAL IC protocol

Prior to each IC measurement, all participants will receive detailed preparatory oral and printed information. Participants will be asked to abstain from food, alcohol and any calorie-containing beverages for at least 8 hours and refrain from vigorous physical activity for at least 12 hours prior to the measurement. Participants will arrive between 10 and 12 hours after an overnight fast before undergoing the IC test. Participants will be asked to confirm their fasting duration and will be reminded to follow preparatory procedures prior to next visit.

The ECAL IC (Metabolic Health Solutions, Australia) used will be calibrated using the 5% carbon dioxide to calibrate gas flow, as per manufacturer specifications. Anthropometric data will be entered into the EC Health software linked up to the ECAL IC. Participant will be lying comfortably in a supine position and elevated at a 45°angle. After 15 min, a nose clip will be applied over the nostrils and participant will be instructed to breathe through a single-use mouthpiece, ensuring a tight seal over it. The test instructor will observe the volumetric flow and ensure that a steady volume and rate of breathing is maintained for between 8 and 10 min. A successful test will be defined as a steady-state achieved with a minimum of five consecutive minutes with less than 10% coefficient of variation in $FEO_2$ (fractional concentration of oxygen in expired air) and $FECO_2$ (fractional concentration of carbon dioxide in expired air).[27] The instructor will stop the IC measurement when this steady-state has been achieved. Should any interruptions occur or failing to achieve a steady-state, a repeat testing will be performed.

### Client report

A client report will be generated using the EC Health software when the IC measurement is complete. A series of questions related to their time-specific metabolism will be completed and a client report generated for participant and clinician in the INT group only. An example of client report can be found in online supplemental appendix.

### Resting energy expenditure

All participants including the SC and INT group will undergo ECAL IC after an overnight fast at baseline, week 2, 4, 8, 12, 16, 20 and 24 visits, using a mouthpiece while in a supine and rested state. The ECAL IC will be calibrated as described within the ECAL IC protocol section. All ECAL IC testing will be conducted in the morning in a temperature-neutral environment with participants lying in a comfortable supine position. During the measurement duration of between 8 and 10 min, participants will be asked to remain as relaxed as possible without falling asleep and instructed not to speak or fidget. The $VO_2$ and $VCO_2$ will be continuously measured for 8–10 min. After discarding the first 2 min of data, REE will be calculated as the lowest consecutive 5 min average value, provided that the coefficient of variation within that 5 min interval is <10%. The ECAL IC will calculate the REE based on the Weir equation.[28] This information will be used by the dietitian to formulate a dietary plan and make recommendations of an energy-restricted diet for participants randomised to the INT group. In contrast, participants and dietitians providing care within the SC group will not receive this information but dietary estimate of TDEE will be based on the Harris-Benedict predictive EE equation.

### Respiratory quotient

Similarly, the ECAL IC testing conducted at baseline, week 2, 4, 8, 12, 16, 20 and 24 visits, will generate an RQ value alongside the REE. The IC will analyse the respiratory

gas exchange of the participants to determine VO2 and VCO2 and the ratio of VCO2 to VO2 (VCO2/VO2), will determine the RQ. RQ will be utilised as an indicator of measurement adequacy and of substrate oxidation. Based on the gas exchange dataset, the ECAL IC will generate information on carbohydrate oxidation (%) and fat oxidation (%). This information will be used by the dietitian to formulate a dietary plan and make recommendations of an energy-restricted diet for participants randomised to the INT group.

## Dietary analysis

Participants will be provided instructions on completing the 3-day food diary. Participants will be provided with cups, spoons or use a food scale to accurately determine food portion sizes. Participants will be asked to record estimate volumes and portion sizes. Issues regarding estimating portion sizes and measuring foods will be discussed. Participants will be informed to record 2 week days and 1 weekend day before their next study visit and were told to provide exact brand names where possible. Participants will be encouraged to select days when their normal routine was least likely to be disrupted and were instructed to record meals eaten away from the home to the best of their ability. All food diaries will be checked by the study dietitians for missing data or obvious errors in recording. Adherence to the energy-restricted dietary recommendations (30% below TDEE) will be assessed using the food diary 3×24 hours dietary recalls together with the dietician at study visit (baseline, week 4, 8, 12, 16, 20 and 24). The threshold of compliance to recommended dietary restriction based on self-reported food diary will be set at >80%. All participants will be asked to return the food diary and provided a reminder to bring along the subsequent food diary at the following visit.

## Activity energy expenditure

Physical activity will be evaluated based on the self-reported IPAQ long form[29] at baseline, week 12 and week 24. Participants will be asked to record the duration and frequency of mild-intensity, moderate-intensity and vigorous-intensity activity levels within the past 7 days. The hours spent in sleep and mild-activity, moderate-activity and vigorous-activity were multiplied by respective metabolic equivalents (METs), summed and finally expressed as total MET-h/week. The data on physical activity from self-reported IPAQ captured will be used to calculate energy recommendation for weight loss.

## Quality of life assessment

Assessment of quality of life outcome measures will be conducted via the IWQoL-Lite and EQ-5D-5L questionnaires. Participants will complete these questionnaires at baseline and final visit (week 24).

## Meal test

All participants will be offered the option of participating in a meal test consisting of 35% of the participant's calculated 24-hour energy requirement in the form of the Ensure Plus high-caloric milkshake. After an overnight fast of at least 8 hours, participants will be offered Ensure Plus milkshake which consists of a 220mls of milkshake style oral nutritional supplement with energy content of 1.5 kcal/mL. Ensure Plus consist of 53.8% calories from carbohydrate, 29.5% from fat and 16.7% from protein. Participants will be given up to 5 min to consume these high-caloric milkshakes worth 35% of their total daily energy needs. The majority of participants will be having to consume less than 500 mL of high-caloric milkshake. Blood samples will be taken to determine baseline concentrations of plasma glucose, insulin and appetite-related gut hormones including GLP-1, GIP and PYY. Blood samples will then be taken at 15, 30, 60, 90 and 120 min after the meal ingestion.

## Glycaemic variability

We will investigate the secondary impact of weight loss on glycaemic variability in a subset of obese individuals with pre-diabetes and compare the changes in mean amplitude of glycaemic excursion using continuous glucose monitoring. We aim to recruit up to 15 participants with pre-diabetes, defined as glycated haemoglobin of 42–47 mmol/mol and an equal representation of another 15 patients without diabetes to undergo this sub-group study.

## Data management

All study participants will be given a unique identifier which will be used to identify their electronic and paper-trail data and biochemical lab samples. The database will be stored on a password-protected computer, accessible to the study researchers only, containing participant identifiers and their associated unique identifier. Paper-based data will be stored securely at the Research & Development Centre, Clinical Sciences Building, University Hospital Aintree, Liverpool, UK for 15 years, after which it may be destroyed. Case report forms will be utilised to compile and collect data at all study visits. Before an analysis, data will be compared between files to ensure accuracy of transcription. Biological samples including blood tests will be stored in a secure swipe card access secured −80°C freezer, with a temperature logbook and alarm system that alerts a staff member should temperature rise above a predetermined set range. Every sample stored in the freezer will be recorded in the logbook of biological samples. Samples will be stored for up to 15 years from collection date and disposed of accordingly after that time.

## Data monitoring

The study will not have a formal data monitoring committee as adverse events of treatments are well known within weight loss INT within a specialist healthcare setting. Any unexpected serious adverse events or outcomes (such as incapacitation or death) will be discussed by the trial management committee (identical to the authors of this protocol). Furthermore, the trial management committee will monitor recruitment,

treatment and attrition rates and any concerns related to the study.

## Protocol deviations

Deviations from proposed study protocol will be communicated to the study sponsor at University of Liverpool/Liverpool Joint Research Office and the Research & Development Unit based at University Hospital Aintree, Liverpool.

## Adverse events

Adverse events will be captured onto the case report forms and will be reported to the University of Liverpool Liverpool Joint Research Office and the Research & Development Unit based at University Hospital Aintree, Liverpool. Adverse events that lead to participant withdrawals will be reported in any future publications. We do not intend to formally analyse adverse events.

## STATISTICAL ANALYSIS PLAN

Statistical analysis will be performed using IBM SPSS Statistics for Windows V.24.0 or later. All variables will be checked for plausibility and missing values. Outcome data will be presented as mean (SD) for continuous variables and counts (percentages) for categorical variables. The characteristics of completers will be tested using independent t-tests for between group analyses and paired t-test for within group analyses. A non-parametric counterpart will be used if data are not normally distributed. $\chi^2$ tests will be used for categorical variables. The primary analysis will use linear mixed models to assess impact of INT (EE information plus SC vs SC alone) on the net change in body weight (baseline, week 24), in absolute terms (kg) between both groups. The covariates age, sex and BMI will also be included in the primary analysis model. Both intention-to-treat (ITT) and per-protocol analyses (for those who achieve a minimum of five out of the nine study visits with completion of IC tests as defined in the 'study INT') will be completed. Where main effects are identified, Bonferroni post hoc tests will be performed to identify significant differences between means (a significant p value set at <0.05). While the ITT analysis will be the main analysis, the per-protocol analysis will allow us to determine that the true efficacy of the INT for participants who strictly adhered to the protocol.

The secondary analyses will include change of body composition (body fat %, FFM), change in REE, change in RQ with time (treated as categorical with levels at baseline, 4, 8, 12 and 24 weeks), and treatment-by-time interaction. We did not undertake formal sample size calculation for secondary analysis as the study was powered to the primary endpoint only. As a result, this will provide data as exploratory analysis only. Type I error will be modified by using Bonferroni adjustment or the equivalent non-parametric test. Age, BMI, FFM, obesity-related comorbidities (King's Obesity Staging Criteria), quality of life outcomes and glycaemic variability will be examined to determine whether they contribute significantly to the models. If a covariate is significant, a term will be added to the model to adjust for the effects and two-way interactions with time and treatment will also be examined. If these interactions are also significant they will be similarly adjusted for in the model using analysis of covariance.

## Data access

There are no contractual agreements that require the data from this trial to be shared.

## ETHICS AND DISSEMINATION

Ethics approval was obtained from the Health Research Authority and the North West Research Ethics Committee (18/NW/0645) and is conducted in accordance with the Declaration of Helsinki and Good Clinical Practice. Participants will also receive a copy of their individual results including anthropometric data, body weight change and results of EE and RQ from ECAL IC tests. They will also be provided with a lay summary of overall results of the study. Findings from this study will be disseminated at relevant scientific conferences and the data published in research manuscripts in peer-reviewed journals.

## DISCUSSION

The study goal is to assess the impact of utilising REE and RQ from IC as an adjunctive management for weight loss. The role of utilising EE information to influence behaviour has yet to be evaluated within a real-world pragmatic RCT. Further, the results will help advance existing evidence about the relationship between RQ, REE and change in body weight.

Several experimental studies evaluating change in REE as a response to overfeeding (ie, 40%–70% increase from baseline energy requirements) suggested that short-term (2–8 weeks) overfeeding produces less weight gain than expected from the controlled increase in intake, due to greater increase in REE.[30–34] Such compensatory increase in energy intake lead to weight increases, of which 55%–67% constitutes fat mass gain.[30 31 35 36] Conversely, underfeeding studies reveal that in short-term, intentional weight loss leads to decrease in EE beyond predicted values.[37–39] Decrease in REE is disproportionate to weight measured per kilogram against FFM after weight loss. The regression analysis of change in REE against the change in FFM reveals that decline in REE tended to be greater than accounted for by loss of FFM. Such overcompensatory metabolic changes act to oppose further weight change.[40]

Variability in substrate oxidation between individuals may be an inherent variability that contributes towards weight regain and fat storage. Variability of macronutrient composition intake between participants may influence the measured RQ and influence the results associated with fat vs carbohydrate oxidation.[41] Several studies have suggested macronutrient alterations could

also influence appetite regulation.[42 43] Further, individual differences in circulating insulin levels or sensitivity may confound the reported associations between RQ, REE and fat mass.[14 44 45] We have taken this into account and have therefore excluded patients with diabetes mellitus or metabolic disorders.

IC use has been validated through several detailed experimental in vitro and in vivo studies, demonstrating accuracy of measuring EE when compared with the mass spectrometry and Douglas Bag method.[46] Given that previous studies on other commercial metabolic monitors showed variations in gas exchange rates,[47] leading to secondary metabolic results variability, participants will be invited to strictly adhere to the protocol requirements and all measurements performed under controlled and steady-state experimental setting.

## Strengths

There are several strengths to the study methodology, in particular the randomised controlled trial design. This will be the first RCT to assess whether providing EE and RQ from ECAL IC will influence the outcome of weight loss INT when integrated with communication between the clinician and the patient. The study is powered statistically to evaluate weight change within the primary analysis model. The study has the potential to expand our understanding of how EE information can be integrated into communication and behaviour modification aspects of weight management clinics, while providing assessment of detailed secondary outcome measures including but not limited to body composition, EE, substrate oxidation, quality of life outcome measures. Further, the secondary analysis provides a comparison between the measured REE from IC with that of predicted REE from Harris-Benedict equation which serves to enhance our understanding of accuracy and practicality of use of ECAL IC to help determine weight change in obesity. One of the challenges with any weight management study is to maintain high levels of motivation and engagement. We have been successful in recruiting large number of participants for similar weight loss INT trials and plan to provide support for participants with routine appointments with the dietician and multidisciplinary weight management team.

## Limitations

While the ECAL IC is purpose-built as a portable, convenient and practical device designed to capture EE information in users over a relatively brief time period, ECAL IC provides a wider degree of variation and greater limits of acceptability in the repeatability of the REE, RQ and substrate oxidation measures during steady state when the device is compared against other IC machines using Deltatrac as the standard reference method.[8] In addition, the measured REE will be influenced by age, sex, body size and variation in individual physical activity levels.[48 49] Under strict controlled research lab environments with overnight fasting, the EE information obtained are accurate within the time constraints of those settings (steady

state & fasting) only, but may not reflect the day-to-day fluctuations in the assessment of energy cost of participants due to energy cost of physical activity, change in energy intake and macronutrient dietary composition.[50] The utility of IPAQ as a self-reported indicator to determine physical activity level in people with obesity showed a weaker correlation, and tended to overestimate activity levels when compared against reference standards using an accelerometer or pedometer.[51] A recent systematic review reported that increased exercise and activity levels, including short-term moderate-intensity to high-intensity exercise training was associated with only modest changes in body composition.[52] The reliance on bioelectrical impedance analysis to capture data on body composition may result in a wide degree of variation in capturing data on fat-mass and FFM when compared against the use of body densitometry. This study is designed to evaluate the impact of INT during the 24-week duration but does not specifically evaluate the impact of providing EE information during the weight maintenance phase after the initial weight loss.

This 6-month RCT will provide practical feedback on acceptability and applicability of EE and RQ measured from the portable ECAL IC to facilitate weight loss strategies. This may lead to incorporating strategies for weight loss using EE and RQ as basis of dietetic advice and recommendations which will inform the design of future weight management trials.

**Acknowledgements** Professor JW and UA were primary supervisors for JL in conduct of this study contributing towards completion of PhD.

**Contributors** The study chief investigators JL, UA, DC and JW were responsible for creating the research question, design of the study, obtaining ethical approval, the acquisition of funding and oversight to study implementation. All authors were responsible for the drafting of this manuscript.

**Funding** This work was funded by the University of Liverpool (UoL001379). The ECAL indirect calorimeter was provided and maintained by the manufacturer Metabolic Health Solutions, Australia.

**Disclaimer** The funding source and funding bodies did not have any input into the design of the study, the collection or analysis of data, the preparation of this manuscript or the decision to submit this manuscript for publication.

**Competing interests** JW reports personal fees, grants, and consultancy fees paid to his institution from AstraZeneca, Novo Nordisk, and Takeda, personal fees and consultancy fees paid to his institution from Boehringer Ingelheim, Janssen, Napp, Mundipharma, and Sanofi, and consultancy fees paid to his institution from Lilly, Rhythm Pharmaceuticals and Wilmington Healthcare outside of the submitted work. JL, UA and DC has no competing interests to declare.

**Patient consent for publication** Obtained.

**Provenance and peer review** Not commissioned; externally peer reviewed.

**ORCID iD**
Jonathan Lim http://orcid.org/0000-0003-3682-8910

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
