## [Reviewer comments · BMJ Open]

ARTICLE DETAILS

TITLE (PROVISIONAL)	Design of a Randomised Controlled Trial: Does Indirect Calorimetry Energy Information Influence Weight Loss in Obesity?
AUTHORS	Lim, Jonathan; Alam, Uazman; Cuthbertson, Daniel; Wilding, John

VERSION 1 – REVIEW

REVIEWER	Florêncio Diniz-Sousa University of Porto, Faculty of Sport
REVIEW RETURNED	30-Oct-2020

GENERAL COMMENTS	This is an important study that aimed to evaluate whether energy expenditure information [resting energy expenditure (RRE) and respiratory quotient (RQ)] obtained from indirect calorimetry might help subjects with overweight and obesity enrolled on a structured intensive lifestyle intervention to obtain additional benefits on weight loss. To that, Lim and colleagues proposed a randomized controlled trial protocol, in which is presented the rationale for the study as well as the methodological description included in the research plan. The main difficulty that I had it was to find the methodological description required on SPIRIT guidelines. While the information of some SPIRIT guidelines items is not described anywhere, others are sometimes described in the main manuscript or in the supplementary material (ECAL_study_protocol_v1.2_24-07-2018_AUH), which difficult the understanding of the methodological protocol. I recommend that all SPIRIT guideline items should be in the main manuscript, even if described succinctly, or otherwise clearly indicating where SPIRIT guideline items are described among all the documents (do not forget to add page numbers to the documents). My comments are as follows: Major points Comment #1. Page 8, Line 5. The validity of ECAL was not proved in the study of Kennedy et al. included as reference. Kennedy et al. showed that proportional bias and wide limits of agreement were observed on RMR of ECAL when compared to those results obtained from Deltatrac (standard reference machine). The statement regarding the validity of ECAL should be reformulated. Comment #2. Page 8, Line 8. The reliability of ECAL is considerably worse compared to Deltatrac. This is specially concerning inasmuch as data from steady-state period also revealed high variability [Kennedy et al. described "Analysis of steady-state data revealed that many of the ECAL sessions (n 10 VO₂, n 11 VCO₂) were not within the target 20 % CV."]. Again, statements regarding ECAL reliability should be made with caution.
---

	Comment #3. Page 10, Line 45. Is there some interim analysis pre-specified? If yes, what the reasons to be performed and at what timepoint in the study will this be done? Comment #4. Page 12, Line 35. The following sentence should not be in the Blinding description “Both participants and researchers will be masked to the ECAL results in the SC arm.” This information is not related to masking type, but to the intervention and assessments itself and therefore can be misleading. In fact, this statement is already included in the description of the standard care group. The identification about who was masked has already been perfectly described in the following sentences “It will not be possible to ‘mask’ researchers or participants to group allocation. Those responsible for analysis will be masked to group allocation”. This should also be clarified on clinical.trials (NCT03638895) on “Masking” and “Masking Description” fields. Comment #5. Page 14, Line 15. It is not clear if weight change comparison will be performed with the absolute values (kg) or based on percentage change. Please, describe the outcomes according to SPIRIT guideline item #12. Comment #6. Page 13, Line 28. Page 14, Line 5; Page 23, Line 4; The description of the assessments that will be carried out over time is insufficient. Please, briefly describe each assessment, its validity on subjects with overweight and obesity and the outcomes that will be utilized. An alternative solution can be found on Comment #7. Comment #7. Page 14, Line 33. The outcomes are not listed in the Appendix 1 as mentioned. Please, include as table the same information as described on supplementary material - ECAL_study_protocol_v1.2_24-07-2018_AUH, page 53, line 18. Comment #8. Page 16, Line 37. Please insert a reference reporting the results from SWMS retrospective dataset that was used to calculate sample size. Comment #9. Page 16, Line 45. Please include the time point for what was defined the analysis of between-group differences. Comment #10. Page 17, Line 19. Why were fat-free mass and resting energy expenditure included in the base model? The referred model seems to be more related with secondary objectives (e.g., “To determine whether REE controlled for %FFM is a predictor for weight loss”) than with the primary objective. Perhaps is better to clarify the different statistical analysis approaches that will be used for each objective. Comment #11. Page 17, Line 19. What are the fixed and random factors? Comment #12. Page 17. It is not clear if between-group comparisons of secondary outcomes will be only performed at 24-weeks post-intervention or in more time points of the follow-up. If multiple between-group comparisons throughout the follow-up occur (e.g., at 8, 16, 24-weeks), how will Type I error rate inflation be corrected? Comment #13. Page 17. Is it treatment effect (e.g., weight between-group comparisons at 24-weeks) net of any baseline differences
--	---

	between groups (even if these differences are not significant)? Comment #14. Page 17, Line 6. Between-group comparisons at randomization (baseline moment) are not recommended (more information on CONSORT item 15). Comment #15. Page 18, Line 3. What macro will be used? One already described in the literature? Please describe the methods that will be used to obtain the p-values and confidence intervals. Comment #16. Page 18, Line 5. Please describe how sample size calculation was made for mediation analyses? Comment #17. Page 18, Line 8. What is a medium sized mediation effect? Which effect-size measures for mediation analyses will be used? Please insert the mediation analyses scale that will be utilized or insert as reference a work defining mediation effect size scale. Comment #18. Page 18, Line 21. According to CONSORT item 8b and 12b, as well other organizations (CPMP/EWP/2863/99, 2003), the randomization processes influenced by covariates, in these case BMI category and sex, should include these covariates in the primary model. Comment #19. Page 5, Line 29; Page 17, Line 31 and Page 20, Line 31. Throughout the text was mentioned that per protocol analysis would be performed, but no methodological description was performed on the entire document. Comment #20. Page 32-37. The information of the following items of SPIRIT guidelines is not clear or is missing assuming the page where the description was indicated.  • Page 32, Line 46. The information of item #2b is not clear on page number 8. • Page 32, Line 50. The information of item #3 is not clear on page number 8. • Page 32, Line 52. The information of item #4 is not clear on page number 8. • Page 32, Line 55. The information item #5a is not clear on page number 8. • Page 33, Line 1. The information of item #5b is not clear on page number 1. • Page 33, Line 6. The information on item #5c is not clear on page number 8. • Page 33, Line 14. The information on item #5d is not clear on page number 8. • Page 34, Line 21. The information on item #12 is not clear on page number 11. • Page 35, Line 27. The information on item #18a is not clear on page number 14. • Page 35, Line 39. The information on item #18b is not clear on page number 14. • Page 35, Line 45. The information on item #19 is not clear on page number 14. • Page 36, Line 10. The information on item #21a is not clear on page number 17. • Page 36, Line 20. The information on item #21b is not clear on page number 17. • Page 36, Line 25. The information on item #22 is not clear on page number 17.
--	--

- Page 36, Line 30. The information on item #23 is not clear on page number 17.
- Page 36, Line 54. The information on item #26b is not clear on page number 8.
- Page 37, Line 1. The information on item #27 is not clear on page number 8.
- Page 37, Line 10. The information on item #29 is not clear on page number 17.
- Page 37, Line 15. The information on item #30 is not clear on page number 17.
- Page 37, Line 19. The information on item #31a is not clear on page number 17.
- Page 37, Line 27. The information on item #31b is not clear on page number 17.
- Page 37, Line 31. The information on item #31c is not clear on page number 17.
- Page 37, Line 37. The information on item #32 is not clear on page number 17.
- Page 37, Line 41. The information on item #33 is not clear on page number 17.

Minor points

Comment #21. Page 7; Line 13. The link indicated on the reference is not directly related to what was mentioned. Please replace with the following link -
https://www.who.int/nmh/events/ncd_action_plan/en/

Comment #22. Page 7; Line 29. Please replace “increased” by “increased”.

Comment #23. Page 7; Line 54. Why did you include the work of Macdonald-Clarke et al. as reference?

Comment #24. Page 10, Line 57. Please update the dates in which recruitment phase will take place.

Comment #25. Page 13, Line 47 and Page 15, Line 38. In which appendix is presented the client report example?

Comment #26. Page 14, Line 45. How long should participants refrain from vigorous physical activity? Although this information is indicated in the supplementary material (ECAL_study_protocol_v1.2_24-07-2018_AUH) this should be referred in the main manuscript.

Comment #27. Page 15, Line 15. As referred on Comment #2, ECLA limitations might bring some difficulties to achieve the defined steady-state with a minimum of 5 consecutive minutes with less than 10% coefficient of variation in FEO₂ and FECO₂. Test repetition is of course a solution, but might become a time-consuming evaluation.

Comment #28. Page 20, Line 10. The indirect calorimetry acronym has already been referred. Please review the use of all acronyms throughout the document.

Comment #29. Page 23, Line 47. One of the aspects that most surprised me in the study design was the little relevance that was

	given to the energy expenditure measurement from daily physical activity. In weight loss interventions it is essential to accurately assess physical activity energy expenditure, alongside resting energy expenditure, especially in programs that propose a more active lifestyle. The utilization of the international physical activity questionnaire (IPAQ) to measure physical activity energy expenditure in subjects with overweight and obesity might be considered a limitation in future publications.
--	---

REVIEWER	Michaela Noreik Nuffield Department of Primary Care Health Sciences, University of Oxford, UK
REVIEW RETURNED	29-Dec-2020

GENERAL COMMENTS	This is an interesting research protocol looking at whether having information on resting energy expenditure (REE) and respiratory quotient (RQ) can improve weight loss when used in the communication between clinician and patient. Although measuring REE in the context of weight loss is not new, it will be interesting to see whether regular inclusion in the counselling with a clinician is more effective than following a weight loss programme alone. In a sub-group glucose levels will be monitored using a continuous glucose monitor to assess glycaemic variability. In another sub-group participants will consume a high-calorie drink and plasma glucose, insulin and appetite hormones will be monitored to look at a possible change in relation to weight loss. Pg. 9 row 3: In the paragraph leading up to this sentence you refer to studies looking at an association between RQ and weight change. But in this sentence you talk about RQ being a valuable indicator of substrate oxidation during an energy-balanced state. But does this also hold for a non-balanced state? If yes, you should state it here with the respective references. If not, the question is whether the statement is true for your study population. Pg. 13 row 10: The exclusion criteria are listed in table one and would not need to be reported double. For clarity it might be good to either add them in the text and delete the table or just refer to the table and delete the list of exclusion criteria in the text. Pg. 13 row 52: It would be good to be more clear about the fact that the intervention group not only receives the results from IC but also have a dietary plan based on the measurement whereas the control group receives a diet based on H&B, which supposedly is less precise. It is nicely presented in the flow-diagram but I think there should be additional information in the methods part. Maybe add another sentence on pg. 13 row 38, so it is clear that they not only receive the report mentioned here but also a dietary plan based on the measurement. Pg. 16 row 45: I think you should move the information that the target is for participants to consume 35% of their daily energy need to the beginning as I was confused as to why you stated '...in a meal test based upon their calculated total daily energy requirement.' You later give the reason but I think it would be easier for the reader to have the information right at the beginning. It would also be good to know how much time participants have to drink the shakes as these high-caloric milkshakes worth 35% of energy need could be challenging to consume in a short period of time. Pg. 16 row 54: The calorie % do not add up to 100%. Do you mean % of average daily energy need? Pg. 17 row 29 'Only the participants with prediabetes, defined as glycated haemoglobin HbA1c of 42 – 47 mmol/mol will be recruited
--

	for this sub- group study.’ But the protocol states ‘We aim to recruit 15 participants with prediabetes and an equal representation of another 15 patients who are non-diabetic to undergo recording with continuous glucose monitoring (CGM) sensor.’ Could you please clarify why this was changed? Pg. 18 row 6: It looks like the word ‘versus’ needs to be deleted. Pg. 21 paragraph starting row 29: I think this paragraph needs to be more specific. Hopefully every study has a detailed statistical analysis plan and also detailed secondary outcomes. The study looks at some interesting aspects. This should be reflected in this final paragraph.
--	---

VERSION 1 – AUTHOR RESPONSE

Reviewer: 1

Dr. Florêncio Diniz-Sousa, Universidade do Porto

Comments to the Author:

This is an important study that aimed to evaluate whether energy expenditure information [resting energy expenditure (RRE) and respiratory quotient (RQ)] obtained from indirect calorimetry might help subjects with overweight and obesity enrolled on a structured intensive lifestyle intervention to obtain additional benefits on weight loss. To that, Lim and colleagues proposed a randomized controlled trial protocol, in which is presented the rational for the study as well as the methodological description included in the research plan.

We thank the reviewer 1 for the constructive and positive feedback. We welcome the feedback and have made specific changes to respond to the comments by Reviewer 1.

The main difficulty that I had it was to find the methodological description required on SPIRIT guidelines. While the information of some SPIRIT guidelines items is not described anywhere, others are sometimes described in the main manuscript or in the supplementary material (ECAL_study_protocol_v1.2_24-07-2018_AUH), which difficult the understanding of the methodological protocol. I recommend that all SPIRIT guideline items should be in the main manuscript, even if described succinctly, or otherwise clearly indicating where SPIRIT guideline items are described among all the documents (do not forget to add page numbers to the documents).

We thank the reviewer 1 for the constructive feedback. We acknowledge that there were gaps that were not adequately covered in the original manuscript submission based on all the items included in the SPIRIT guidelines. We have included additional details and verified that all items on the SPIRIT guidelines are now included in the main revised manuscript. The specific response and changes to each of these points can be found in the section below.

My comments are as follows:

Major points

Comment #1. Page 8, Line 5. The validity of ECAL was not proved in the study of Kennedy et al. included as reference. Kennedy et al. showed that proportional bias and wide limits of agreement were observed on RMR of ECAL when compared to those results obtained from Deltatrac (standard reference machine). The statement regarding the validity of ECAL should be reformulated.

We have amended the statement to reflect a more accurate representation. The amended statement reads as follows (page 5):

ECAL IC has been compared against other IC including the GEM and DeltaTrac. Kennedy et al. (8) reported that measures of VO₂, RMR, RQ, carbohydrate oxidation, and fat oxidation showed greatest variation on the ECAL IC as compared against Deltatrac (as the standard reference device). The mean difference in RMR measures collected on ECAL and Deltatrac showed wide limits of agreement (lower 95% limit of agreement was -2562 kJ/d and upper 95% limit of agreement was 3480 kJ/d). A greater proportional bias was observed between measured RMR of ECAL against Deltatrac suggesting that at higher RMR the difference between the two devices was greater but was acceptable for repeat RMR measures between individuals. In terms of repeatability of measures of ECAL IC (intra-machine variability), Kennedy et al. reported that no significant differences were found between repeated measures of VCO₂, RMR, RQ and substrate oxidation measures. However, greatest bias occurred within the ECAL with a mean difference of 475 ± 1083 kJ/d and wide limits of agreement (-2641 and 1691 kJ/d). Comparing between devices, coefficient of variance was 4 (±5.3)% on the Deltatrac, 4.9 (±4.5)% on the GEM and 11.2 (±12.1)% on the ECAL (8).

Comment #2. Page 8, Line 8. The reliability of ECAL is considerably worse compared to Deltatrac. This is specially concerning inasmuch as data from steady-state period also revealed high variability [Kennedy et al. described “Analysis of steady-state data revealed that many of the ECAL sessions (n 10 VO₂, n 11 VCO₂) were not within the target 20 % CV.”]. Again, statements regarding ECAL reliability should be made with caution.

We have amended the statement to reflect an accurate representation of the observed proportional bias and wide limits of agreement using the ECAL IC as compared to Deltatrac.

The revised statement is on Page 5. Similar to the response to comment #1.

Comment #3. Page 10, Line 45. Is there some interim analysis pre-specified? If yes, what the reasons to be performed and at what time point in the study will this be done?

We have not specified and do not currently intend to perform an interim analysis.

Comment #4. Page 12, Line 35. The following sentence should not be in the Blinding description “Both participants and researchers will be masked to the ECAL results in the SC arm.” This information is not related to masking type, but to the intervention and assessments itself and therefore can be misleading. In fact, this statement is already included in the description of the standard care group. The identification about who was masked has already been perfectly described in the following sentences “It will not be possible to ‘mask’ researchers or participants to group allocation. Those responsible for analysis will be masked to group allocation”. This should also be clarified on clinical.trials (NCT03638895) on “Masking” and “Masking Description” fields.

We have removed the misleading description as pointed out by Reviewer 1.

To improve the systematic flow in the manuscript, we have removed the sentence as requested. We have moved the “Blinding” description to the section titled “Randomisation, allocation, concealment and sequence generation” on page 10.

The revised statement can be found on page 10 as follows:

“It will not be possible to ‘mask’ researchers or participants to group allocation. However, those responsible for analysis will be masked to group allocation. A research assistant who will not be involved in the enrolment, assessment or allocation of participants will pack and sequence pre-packed envelopes with the group allocation. Only after the study investigator/ sub-investigator reviews the eligibility based on health records, laboratory results and concomitant medications, and confirms

eligibility to proceed, will the envelope be opened and details of the particular study group will be revealed to the participant.

We have amended the information on Clinicaltrials.gov on Masking and Masking Description fields.

Comment #5. Page 14, Line 15. It is not clear if weight change comparison will be performed with the absolute values (kg) or based on percentage change. Please, describe the outcomes according to SPIRIT guideline item #12.

We have added and specified the weight change in absolute values (kg). The paragraph for primary outcome now reads as follows (page 13):

The primary outcome is the difference in change of weight, in absolute value (kg), between participants in the INT group (EE information plus SC) versus the SC group at baseline and 24-weeks after randomisation.

Comment #6. Page 13, Line 28. Page 14, Line 5; Page 23, Line 4;

The description of the assessments that will be carried out over time is insufficient. Please, briefly describe each assessment, its validity on subjects with overweight and obesity and the outcomes that will be utilized. An alternative solution can be found on Comment #7.

Thank you for this comment. We have elaborated and provided more detailed description of the assessments that will be carried out.

We have added the following sections highlighted in RED in the revised manuscript:

Page 14 we have added a new section under sub-heading “Anthropometry”

Page 14 we have added a new section under sub-Heading “Biochemical measures”. We have detailed information regarding storage of blood samples.

Page 16 we have added a further sub-heading “Resting Energy Expenditure” to provide a more detailed description of each assessment

Page 16 we have added a further sub-heading “Respiratory Quotient” to provide a more detailed description of measured RQ

Page 17 we have added a new sub-Heading “Dietary Analysis” to provide a more detailed description of how we will evaluate the dietary intake of participants and record of compliance to recommended energy restrictions.

Page 17 we have added a new Sub-heading “Activity Energy Expenditure” to provide more details on how activity EE will be evaluated.

Page 18 we have added a new sub-heading “Quality of Life Assessment”

Each of these sections are related to the complete checklist of of study outcomes and dependent variables summarised in Table 2 (page 29).

Comment #7. Page 14, Line 33. The outcomes are not listed in the Appendix 1 as mentioned. Please, include as table the same information as described on supplementary material - ECAL_study_protocol_v1.2_24-07-2018_AUH, page 53, line 18.

As per the response to Comment #6 above, we have provided further details on each outcome as specified as listed in the Appendix 1 of the study protocol. The complete list of outcomes is summarised in Table 2 (page 30)

Comment #8. Page 16, Line 37. Please insert a reference reporting the results from SWMS retrospective dataset that was used to calculate sample size.

We have included the reference from the SWMS retrospective dataset from reference: (page 11) "Sample Size Calculation" – The sample size calculation was based upon retrospective dataset from SWMS.

Reference added: Steele T, Narayanan RP, James M, James J, Mazey N, Wilding JPH. Evaluation of Aintree LOSS, a community-based, multidisciplinary weight management service: outcomes and predictors of engagement. Clin Obes. 2017;7(6):368-76.

Comment #9. Page 16, Line 45. Please include the time point for what was defined the analysis of between-group differences.

We have included the time point at 24 weeks for analysis of between-group differences (page 20 & 21).

Comment #10. Page 17, Line 19. Why were fat-free mass and resting energy expenditure included in the base model? The referred model seems to be more related with secondary objectives (e.g., "To determine whether REE controlled for %FFM is a predictor for weight loss") than with the primary objective. Perhaps is better to clarify the different statistical analysis approaches that will be used for each objective.

Thank you for pointing this out. The revised paragraph now reads as follows: (page 20 & 21). The primary analysis will utilise linear mixed models to assess impact of intervention (EE information plus SC vs. SC alone) on the net change in body weight (baseline, week 24), in absolute terms (kg).

The fat-free mass and REE have been removed from the base model and included in the secondary analysis consistent with the study objectives.

The revised paragraph now reads as follows: (page 21) 'The secondary analyses will include change of body composition (body fat %, fat-free mass), change in REE, change in RQ with time (treated as categorical with levels at baseline, 4, 8, 12 and 24 weeks), and treatment-by-time interaction'

Comment #11. Page 17, Line 19. What are the fixed and random factors?

Thank you for your comment. The revised phrase now reads as follows: We have specified the fixed factors as group allocation. Analysis will also include change in body composition, change in REE and change in RQ with time within the secondary analysis (page 21).

Comment #12. Page 17. It is not clear if between-group comparisons of secondary outcomes will be only performed at 24-weeks post-intervention or in more time points of the follow-up. If multiple between-group comparisons throughout the follow-up occur (e.g., at 8, 16, 24-weeks), how will Type I error rate inflation be corrected?

We intend to perform multiple between-group comparisons at several time points including week 4, 8, 12 and week 24 to evaluate the treatment-by-time changes. We have selected these time points to investigate if changes in secondary outcomes could predict the change in body weight at week 24. We have included the sentence: (page 21) "Type I error will be modified by utilizing Bonferroni adjustment" under the section heading of Statistical Analysis Plan.

Comment #13. Page 17. Is it treatment effect (e.g., weight between-group comparisons at 24-weeks) net of any baseline differences between groups (even if these differences are not significant)?

Thank you asking us to clarify this point. We have specified the net change of baseline differences and week 24 between both groups.

The revised sentence reads as follows:

(page 20 & 21) The primary analysis will utilise linear mixed models to assess impact of intervention (EE information plus SC vs. SC alone) on the net change in body weight (baseline, week 24), in absolute terms (kg) between both groups.

Comment #14. Page 17, Line 6. Between-group comparisons at randomization (baseline moment) are not recommended (more information on CONSORT item 15).

Thank you for highlighting this oversight. We have now removed baseline between-group comparisons as kindly pointed out by Reviewer 1.

Comment #15. Page 18, Line 3. What macro will be used? One already described in the literature? Please describe the methods that will be used to obtain the p-values and confidence intervals.

Thank you for highlighting this. We will utilise the SPSS Statistics for Windows V.24.0 or later. We have removed the sentence “Macro generates bias-corrected 95% confidence intervals around the indirect effect.” given it was an inaccurate representation.

We have revised the paragraph statistical analysis plan in the manuscript (page 21 & 22).

In order to determine significance, we will utilize ANCOVA.

Comment #16. Page 18, Line 5. Please describe how sample size calculation was made for mediation analyses?

Thank you for pointing this out. We have removed the “mediation analysis” in the secondary analysis as this was relatively unclear. Instead, we have specified the intention to perform correlation and multiple linear regression based on the secondary analysis of the dependent variables described in the manuscript (depending on the parametric or non-parametric distribution of the dataset).

For the secondary analysis, there was no formal sample size calculator /was not undertaken to ascertain whether the numbers was adequate for correlation.

We did not undertake formal calculations for sample size for secondary analysis as the study was powered to the primary endpoint only. We have added a sentence (page 21) “We did not undertake formal sample size calculation for secondary analysis as the study was powered to the primary endpoint only. As a result, this will provide data as exploratory analysis only.” In the secondary analysis description. As a result, this was exploratory analysis only.

Comment #17. Page 18, Line 8. What is a medium sized mediation effect? Which effect-size measures for mediation analyses will be used? Please insert the mediation analyses scale that will be utilized or insert as reference a work defining mediation effect size scale.

We will not undertake the analysis of mediation effect. We have removed this in the revised manuscript.

Comment #18. Page 18, Line 21. According to CONSORT item 8b and 12b, as well other organizations (CPMP/EWP/2863/99, 2003), the randomization processes influenced by covariates, in these case BMI category and sex, should include these covariates in the primary model.

We have now specified “the covariates BMI and sex will be included in the primary model”. (page 21)

Comment #19. Page 5, Line 29; Page 17, Line 31 and Page 20, Line 31. Throughout the text was mentioned that per protocol analysis would be performed, but no methodological description was performed on the entire document.

Thank you for pointing this out. We have specified and defined the per protocol analysis within the section statistical analysis plan.

We have amended the sentence to read as follows:

(page 20 row 23) Both intention-to-treat (ITT) and per protocol analyses (for those who achieve a minimum of 5 out of the 9 study visits with completion of IC tests as defined in the ‘study intervention’) will be completed.

Comment #20. Page 32-37. The information of the following items of SPIRIT guidelines is not clear or is missing assuming the page where the description was indicated.

- Page 32, Line 46. The information of item #2b is not clear on page number 8.
- Page 32, Line 50. The information of item #3 is not clear on page number 8.
- Page 32, Line 52. The information of item #4 is not clear on page number 8.
- Page 32, Line 55. The information item #5a is not clear on page number 8.
- Page 33, Line 1. The information of item #5b is not clear on page number 1.
- Page 33, Line 6. The information on item #5c is not clear on page number 8.
- Page 33, Line 14. The information on item #5d is not clear on page number 8.
- Page 34, Line 21. The information on item #12 is not clear on page number 11.
- Page 35, Line 27. The information on item #18a is not clear on page number 14.
- Page 35, Line 39. The information on item #18b is not clear on page number 14.
- Page 35, Line 45. The information on item #19 is not clear on page number 14.
- Page 36, Line 10. The information on item #21a is not clear on page number 17.
- Page 36, Line 20. The information on item #21b is not clear on page number 17.
- Page 36, Line 25. The information on item #22 is not clear on page number 17.
- Page 36, Line 30. The information on item #23 is not clear on page number 17.
- Page 36, Line 54. The information on item #26b is not clear on page number 8.
- Page 37, Line 1. The information on item #27 is not clear on page number 8.
- Page 37, Line 10. The information on item #29 is not clear on page number 17.
- Page 37, Line 15. The information on item #30 is not clear on page number 17.
- Page 37, Line 19. The information on item #31a is not clear on page number 17.
- Page 37, Line 27. The information on item #31b is not clear on page number 17.
- Page 37, Line 31. The information on item #31c is not clear on page number 17.
- Page 37, Line 37. The information on item #32 is not clear on page number 17.
- Page 37, Line 41. The information on item #33 is not clear on page number 17.

We thank the reviewer for listing all the items on SPIRIT checklist that require further clarification. The page numbers that we have listed corresponds to the main manuscript that we submitted to journal. For clarification we have revisited each item on the checklist and reviewed the indicated page.

In response to the requirement of the SPIRIT checklist we have added the following new sub-headings which were highlighted in RED font

On page 19, we have added a new subheading 'Data Management'

On page 19, we have added a new subheading 'Data Monitoring'

On page 20, we have added a new subheading 'Protocol Deviations'

On page 20, we have added a new subheading 'Adverse events'

On page 22, we have added a new subheading 'Data access'

On page 22, we have expanded the section on 'Ethics and dissemination'

- The information of item #2b is not clear on page number 8.

Trial Registration Data set listed on page number 8-9, 19-22.

- Page 32, Line 50. The information of item #3 is not clear on page number 8.

The information on protocol date and version is on page number 8.

- Page 32, Line 52. The information of item #4 is not clear on page number 8.

Source and type of financial, material and other support on page number 8-9, 26-27

- Page 32, Line 55. The information item #5a is not clear on page number 8.

Names affiliations, roles and protocol contributors on page number 26-27

- Page 33, Line 1. The information of item #5b is not clear on page number 1.

Name and contact information for trial sponsor on page number 8-9, 22, 26-27

- Page 33, Line 6. The information on item #5c is not clear on page number 8.

The information on item #5c is on page number 8-9, 19-20, 22, 26-27

- Page 33, Line 14. The information on item #5d is not clear on page number 8.

The information on item #5d is on page number 8-9, 19-20, 22, 26-27

- Page 34, Line 21. The information on item #12 is not clear on page number 11.

The information on item #12 is on page number 13-20, 30-31

- Page 35, Line 27. The information on item #18a is not clear on page number 14.

Further information added on data collection plan. The information on item #18a is on page number 19-22

- Page 35, Line 39. The information on item #18b is not clear on page number 14.

Further information added on data collection plan. The information on item #18b is on page number 19-22

- Page 35, Line 45. The information on item #19 is not clear on page number 14.

Further information added on data management. The information on #19 is on page number 19-22

- Page 36, Line 10. The information on item #21a is not clear on page number 17.

The information

Further information added on data monitoring. There is no formal data monitoring committee for the trial. This is now included in the revised manuscript under section "Data Monitoring". Information can be found on page 20-22

- Page 36, Line 20. The information on item #21b is not clear on page number 17.

Further information added. This is now included on page 20-22

- Page 36, Line 25. The information on item #22 is not clear on page number 17.

Further information added to section "Adverse Events". Information on page 20-22

- Page 36, Line 30. The information on item #23 is not clear on page number 17.

Further information added to section. This information is now on page 19-22

- Page 36, Line 54. The information on item #26b is not clear on page number 8.

Further information regarding consent and collection of participant data and biological specimens have been added. This information can be found in section "Data management" on page 9-10,22

- Page 37, Line 1. The information on item #27 is not clear on page number 8.

Further information regarding how personal information of enrolled participants will be stored have been added under section "Data management" on page 19-22

- Page 37, Line 10. The information on item #29 is not clear on page number 17.

Further information on Data Access have been added to section "Data access" found on page 22.

- Page 37, Line 15. The information on item #30 is not clear on page number 17. Further information on item #30 can be found under “Adverse events” on pages 19-20,22.
- Page 37, Line 19. The information on item #31a is not clear on page number 17. Further information on Dissemination policy have been added. Found on pages 22
- Page 37, Line 27. The information on item #31b is not clear on page number 17. Further information on Dissemination policy been added. Found on pages 22,26-27.
- Page 37, Line 31. The information on item #31c is not clear on page number 17. Further information on Dissemination policy have been added,in section. Found on pages 8-9,22
- Page 37, Line 37. The information on item #32 is not clear on page number 17.

Model consent form

A model consent form is attached as appendix item to the manuscript. – Attached Consent Form in Appendix.

- Page 37, Line 41. The information on item #33 is not clear on page number 17. Further information on item #33 can be found on pages 14,18-19.

Minor points

Comment #21. Page 7; Line 13. The link indicated on the reference is not directly related to what was mentioned. Please replace with the following link -
https://www.who.int/nmh/events/ncd_action_plan/en/

Thank you for raising this issue. The reference was replaced with the link to read as follows:
 World Health Organisation. Global Action for the Prevention and Control of NCDs 2013-2020 2013
 [Available from: https://www.who.int/nmh/events/ncd_action_plan/en/].

Comment #22. Page 7; Line 29. Please replace “Increased” by “increased”.

We have amended as indicated by reviewer. (page 4 row 20)

Comment #23. Page 7; Line 54. Why did you include the work of Macdonald-Clarke et al. as reference?

The reference Macdonald-Clarke was erroneously included as a reference intended to explain how ECAL indirect calorimetry device functioned. The work by Macdonald-Clarke presented preliminary data from measurement of REE by ECAL as compared to the QUARK (standard reference) but we were unable to obtain a full text article containing the full report. Therefore, we have removed the reference.

Comment #24. Page 10, Line 57. Please update the dates in which recruitment phase will take place.

The recruitment phase have been updated which will include dates from February 2019 until February 2022 under the section Recruitment (page 9 row 11).

Comment #25. Page 13, Line 47 and Page 15, Line 38. In which appendix is presented the client report example?

We have included the exact page for the client report example. – Attached Client Report in Appendix.

Comment #26. Page 14, Line 45. How long should participants refrain from vigorous physical activity? Although this information is indicated in the supplementary material (ECAL_study_protocol_v1.2_24-07-2018_AUH) this should be referred in the main manuscript.

We have included this information in the revised manuscript. The sentence now reads as follows: Participants will be asked to refrain from vigorous physical activity for at least 12 hours prior to the measurement. (page 14 row 23)

Comment #27. Page 15, Line 15. As referred on Comment #2, ECAL limitations might bring some difficulties to achieve the defined steady-state with a minimum of 5 consecutive minutes with less than 10% coefficient of variation in FEO₂ and FECO₂. Test repetition is of course a solution, but might become a time-consuming evaluation.

Given the ECAL limitations may pose some difficulties to achieve steady-state testing, we have specified that test repetition would be performed in instances when readings did not reach the steady-state definition.

Comment #28. Page 20, Line 10. The indirect calorimetry acronym has already been referred. Please review the use of all acronyms throughout the document.

Thank you. We have amended as informed and replaced the indirect calorimetry acronym.

Comment #29. Page 23, Line 47. One of the aspects that most surprised me in the study design was the little relevance that was given to the energy expenditure measurement from daily physical activity. In weight loss interventions it is essential to accurately assess physical activity energy expenditure, alongside resting energy expenditure, especially in programs that propose a more active lifestyle. The utilization of the international physical activity questionnaire (IPAQ) to measure physical activity energy expenditure in subjects with overweight and obesity might be considered a limitation in future publications.

We agree with reviewer 1 that in the study design we intended for physical activity measures to be captured using IPAQ. We accept that this would pose a methodological limitation to the accurate determination of energy expenditure measurements but chose in favour of using IPAQ as a proxy for activity energy expenditure to reduce the high burden on participants.

Reviewer: 2

Dr. Michaela Noreik, University of Oxford Department of Primary Care Health Sciences

Comments to the Author:

See enclosed PDF document

Reviewer: 1

Competing interests of Reviewer: None declared

Reviewer: 2

Competing interests of Reviewer: None declared

FORMATTING AMENDMENTS (if any)

Required amendments will be listed here; please include these changes in your revised version:

- Please re-upload your supplementary files in PDF format.

Response to Reviewer 2

This is an interesting research protocol looking at whether having information on resting energy expenditure (REE) and respiratory quotient (RQ) can improve weight loss when used in the communication between clinician and patient.

Although measuring REE in the context of weight loss is not new, it will be interesting to see whether regular inclusion in the counselling with a clinician is more effective than following a weight loss programme alone. In a sub-group glucose levels will be monitored using a continuous glucose monitor to assess glycaemic variability. In another sub- group participants will consume a high-calorie drink and plasma glucose, insulin and appetite hormones will be monitored to look at a possible change in relation to weight loss.

Thank you for taking this opportunity to review this methodology paper. We thank the reviewer for taking an interest evaluating this hypothesis.

Pg. 9 row 3: In the paragraph leading up to this sentence you refer to studies looking at an association between RQ and weight change. But in this sentence you talk about RQ being a valuable indicator of substrate oxidation during an energy-balanced state. But does this also hold for a non-balanced state? If yes, you should state it here with the respective references. If not, the question is whether the statement is true for your study population.

Thank you for the constructive comment. In response we have quoted additional references from studies which reported change in fasting RQ during a non-balanced energy state. We have included the following:

(page 6 row 16) If substrate oxidation was examined in a positive energy balance, i.e. energy intake > expenditure, carbohydrate oxidation increases and fat oxidation decreases (22). Conversely, if in a negative energy balance i.e. energy intake < expenditure, carbohydrate oxidation decreases and fat oxidation increases (23). The predominant factor that influences the accuracy of RQ measure of substrate oxidation was determined by body composition, energy intake and EE (24).

22. Jéquier E, Schutz Y. Long-term measurements of energy expenditure in humans using a respiration chamber. *Am J Clin Nutr.* 1983;38(6):989-98.

23. Saltzman E, Roberts SB. Effects of energy imbalance on energy expenditure and respiratory quotient in young and older men: a summary of data from two metabolic studies. *Aging (Milano).* 1996;8(6):370-8.

24. Hall KD, Bain HL, Chow CC. How adaptations of substrate utilization regulate body composition. *Int J Obes (Lond).* 2007;31(9):1378-83.

Pg. 13 row 10: The exclusion criteria are listed in table one and would not need to be reported double. For clarity it might be good to either add them in the text and delete the table or just refer to the table and delete the list of exclusion criteria in the text.

Thank you for pointing this out. We have removed the exclusion criteria within the text to avoid duplication. The complete list of exclusion criteria can be found on Table 1 on page 27.

Pg. 13 row 52: It would be good to be more clear about the fact that the intervention group not only receives the results from IC but also have a dietary plan based on the measurement whereas the control group receives a diet based on H&B, which supposedly is less precise. It is nicely presented in the flow-diagram but I think there should be additional information in the methods part. Maybe add

another sentence on pg. 13 row 38, so it is clear that they not only receive the report mentioned here but also a dietary plan based on the measurement.

Thank you for pointing this out. We have included another sentence and provide further clarity to the reader. The paragraph now reads as follows:

(page 11 row 12) During the study visit, these results will serve as a reference tool for the dietitian when formulating a dietary plan based on the measured REE to determine total daily EE (TDEE). Further, the RQ data will also be used as an indicator of substrate oxidation to deliver key messages on carbohydrate vs. fat oxidation and to facilitate decision making in energy-restricted dietary recommendations.

Pg. 16 row 45: I think you should move the information that the target is for participants to consume 35% of their daily energy need to the beginning as I was confused as to why you stated ‘...in a meal test based upon their calculated total daily energy requirement.’ You later give the reason but I think it would be easier for the reader to have the information right at the beginning. It would also be good to know how much time participants have to drink the shakes as these high-caloric milkshakes worth 35% of energy need could be challenging to consume in a short period of time.

Thank you for pointing this out. We have moved the information to the top of the paragraph and should now read as follows:

(page 18) “All participants will be offered the option of participating in a meal test consisting of 35% of the participant’s calculated 24-hour energy requirement in the form of the Ensure Plus high-caloric milkshake.... Participants will be given up to 5 minutes to consume these high-caloric milkshakes worth 35% of their total daily energy needs.

Pg. 16 row 54: The calorie % do not add up to 100%. Do you mean % of average daily energy need?

Thank you for pointing this out. We have amended this to reflect the following (page 18) “Ensure Plus consist of 53.8% calories from carbohydrate, 29.5% from fat and 16.7% from protein.”

Pg. 17 row 29 ‘Only the participants with prediabetes, defined as glycated haemoglobin HbA1c of 42 – 47 mmol/mol will be recruited for this sub- group study.’ But the protocol states ‘We aim to recruit 15 participants with prediabetes and an equal representation of another 15 patients who are non-diabetic to undergo recording with continuous glucose monitoring (CGM) sensor.’ Could you please clarify why this was changed?

Thank you for pointing this out. We have amended the sentence which now reads accurately (page 19). We aim to recruit up to 15 participants with prediabetes, defined as glycated haemoglobin HbA1c of 42 – 47 mmol/mol and an equal representation of another 15 non-diabetic patients to undergo this sub-group study.

Pg. 18 row 6: It looks like the word ‘versus’ needs to be deleted.

Thank you for pointing this out. This has been amended in the revised manuscript.

Pg. 21 paragraph starting row 29: I think this paragraph needs to be more specific. Hopefully every study has a detailed statistical analysis plan and also detailed secondary outcomes. The study looks at some interesting aspects. This should be reflected in this final paragraph.

Thank you for pointing this out. We have included further details for this final summary paragraph. We have elaborated further on the strengths and limitations of the study on page 23-25.

VERSION 2 – REVIEW

REVIEWER	Florêncio Diniz-Sousa Faculty of Sport, University of Porto, Portugal
REVIEW RETURNED	12-Feb-2021

GENERAL COMMENTS	All comments have been addressed and I have no further comments to add.
---

REVIEWER	Michaela Noreik Nuffield Department of Primary Care Health Sciences
REVIEW RETURNED	22-Feb-2021

GENERAL COMMENTS	Dear Dr Lim and team, Thank you for your reply to my comments. In my view the paper is complete now. All comments were processed satisfactorily.
---